# ‘How Your Spirit Is Travelling’—Understanding First Nations Peoples’ Experiences of Living Well with and after Cancer

**DOI:** 10.3390/ijerph21060798

**Published:** 2024-06-19

**Authors:** Anneliese de Groot, Bena Brown, Daniel Lindsay, Alana Gall, Nicole Hewlett, Amy Hickman, Gail Garvey

**Affiliations:** 1School of Public Health, The University of Queensland, Brisbane 4006, Australia; bena.brown@health.qld.gov.au (B.B.); daniel.lindsay@qimrberghofer.edu.au (D.L.); alana.gall@scu.edu.au (A.G.); n.hewlett@uq.edu.au (N.H.); amy.hickman@flinders.edu.au (A.H.); g.garvey@uq.edu.au (G.G.); 2Australian Institute of Health Innovation, Macquarie University, Sydney 2113, Australia; 3Southern Queensland Centre of Excellence in Aboriginal and Torres Strait Islander Primary Health Care, Queensland Health, Brisbane 4077, Australia; 4QIMR Berghofer Medical Research Institute, Brisbane 4006, Australia; 5National Centre for Naturopathic Medicine, Southern Cross University, Lismore 2480, Australia; 6College of Medicine and Public Health, Flinders University, Adelaide 5042, Australia

**Keywords:** Indigenous health, health disparities, health equity, cancer, cancer treatment, living well with and after cancer, Aboriginal and Torres Strait Islander, First Nations Peoples

## Abstract

As the number of people living with cancer increases, it is important to understand how people can live well with and after cancer. First Nations people diagnosed with cancer in Australia experience survival disparities relating to health service accessibility and a lack of understanding of cultural needs and lived experiences. This study aimed to amplify the voices of First Nations individuals impacted by cancer and advance the development of a culturally informed care pathway. Indigenist research methodology guided the relational and transformative approach of this study. Participants included varied cancer experts, including First Nations people living well with and after cancer, health professionals, researchers, and policy makers. Data were collected through online Yarning circles and analysed according to an inductive thematic approach. The experience of First Nations people living well with and after cancer is inextricably connected with family. The overall themes encompass hope, family, and culture and the four priority areas included the following: strength-based understanding of cancer, cancer information, access to healthcare and support, and holistic cancer services. Respect for culture is interwoven throughout. Models of survivorship care need to integrate family-centred cancer care to holistically support First Nations people throughout and beyond their cancer journey.

## 1. Introduction

Understanding how people can live well with and after cancer is becoming more important as the number of people living with cancer increases [1]. Many patients experience physical, social, spiritual, and psychological effects after receiving a cancer diagnosis and treatment [2]. Living with cancer can disrupt one’s sense of identity and life purpose, causing stigma and discrimination concerns, and psychological distress amongst family members and carers of those living with cancer [1,3,4]. In addition, interactions with healthcare professionals during diagnosis and treatment deliberations can also further impact a patient’s emotional wellbeing and shape their relationship with healthcare services [5]. The term cancer survivor can mean different things to different people; however, the experience of a person having to navigate these life experiences and challenges after receiving a cancer diagnosis is often referred to as cancer survivorship [3].

While a substantial proportion of the Australian population will survive more than 5 years beyond receiving a cancer diagnosis, there are inequities in outcomes experienced by some groups. Australians experience some of the highest cancer survival rates in the world, with 71% of Australians surviving 5 years post their cancer diagnosis between 2015–2019 [6]. However, First Nations people in Australia experience a lower rate of cancer survival (5-year survival 54% for First Nations people versus 68% for non-Indigenous, 2012–2016) [7]. First Nations people in Australia diagnosed with cancer are more likely to be younger, have greater comorbidities, less participation in cancer screening programs, and are diagnosed with more advanced cancer and receive less cancer treatment compared to non-Indigenous cancer patients [8,9]. The underlying causes of these disparities are complex and whilst many relate to the accessibility of health services, many also relate to a lack of understanding of the cultural needs and the lived experiences of First Nations Peoples living with and after cancer [8,10,11]. For example, Western approaches to health tend to idealise neutrality and objectivity, primarily focus on individuals, and emphasise physical wellbeing [12]. In contrast, First Nations cultures in Australia and around the world view health holistically*,* encompassing not only individual wellbeing but the social, emotional, spiritual, and cultural life of the entire community [12,13,14,15,16].

In the face of these challenges, improving health systems’ abilities to respond to the specific cancer survivorship needs of First Nations Australians is imperative if we are to address these cancer inequities. Approaches to cancer survivorship for First Nations Peoples should be tailored to their specific needs and consider the ongoing impact of colonisation and challenges of accessing culturally safe care. There is some existing evidence to help guide cancer survivorship plans and services for First Nations Peoples. For example, aspects of culture, values, and spirituality have been reported as key aspects that impact on the quality of life of First Nations cancer patients within the first six months of receiving a cancer diagnosis [17]. In addition, many First Nations people living with cancer experience unmet supportive care needs, with physical, psychological, and practical/cultural concerns identified as priority areas [18]. These findings are a starting point to inform policy and service provision priority areas for cancer survivorship for First Nations Peoples diagnosed and living with cancer.

To date, there is limited understanding from the perspectives of First Nations Peoples regarding principles, policies, and service provision related to living well with and after cancer. This study conducted a virtual national roundtable to amplify the voices of First Nations individuals impacted by cancer, with the primary objective of advancing the development of a culturally informed care pathway tailored to the needs of First Nations Peoples living with and after cancer.

## 2. Materials and Methods

The Centre of Research Excellence in Targeted Approaches To Improve Cancer Services for Aboriginal and Torres Strait Islander Australians (TACTICS), a First Nations-led research program funded by Australia’s National Health and Medical Research Council, partnered with the Clinical Oncology Society of Australia (COSA) to conduct a virtual roundtable.

This study was guided by Indigenist methodology as a relational and transformative approach to reflect First Nations Peoples Ways of Knowing, Being, and Doing [19,20]. Relational ontology locates the researchers and participants within a network of reciprocal relationships, whereby each entity is attributed equal importance [19]. Transformative approaches examine underlying power imbalances affecting social, political, and historic contexts within which health is experienced by First Nations Peoples living well with and after cancer [19,20]. With this methodology, First Nations Peoples’ voices have been privileged [19,21,22].

Yarning, an Indigenist research method, guided culturally safe, cooperative and conversational data collection [23]. Yarning is an information-sharing method, whereby each person is accountable for engaging and relating respectfully [19,24,25]. This method redresses power imbalance that can often occur between researcher and participant, enabling a safer environment for participants to share [24]. The current study included six online Yarning circles [24,26,27]. Prompts were used to guide discussion according to the research question [26].

First Nations people living with cancer and/or their family members (18 years of age or older at the time of recruitment), healthcare professionals, researchers, and policy makers were purposively recruited through investigator networks and the Clinical Oncological Society of Australia (COSA) Cancer Survivorship Special Interest Group to participate in the virtual roundtable conducted in March 2021. The healthcare professionals, researchers, and policy makers represented medical, nursing, and allied health professions, and all were involved in cancer care across state and national organisations. Participants were initially emailed a Letter of Invitation outlining the purpose of the roundtable and what participation involved and sought their interest. Snowball sampling was adopted, allowing potential participants to share the invitation to participate with their own networks. Interested potential participants were emailed the Participant Information and Consent Form. Participation was voluntary and written consent was obtained prior to data collection.

The virtual roundtable commenced with presentations on the Australian cancer survivorship landscape and the COSA survivorship model of care to give participants the context for the following discussions [2]. Participants were then allocated to Yarning circles (small groups) via Zoom breakout rooms [28]. The research team purposively allocated participants to the Yarning circles to ensure diversity of participant backgrounds and experiences. Each group was allocated a facilitator (member of the research team) and included 6 to 8 participants, including a minimum of 2 First Nations people affected by cancer. The Yarns aimed to gather the perspectives of participants rather than to achieve data saturation across the breadth of discussion. The facilitators encouraged First Nations people to share their experiences and opinions to ensure their voices were heard and privileged [19].

The online Yarning circle guide included open ended questions developed and refined by the research team, such as “What does survivorship mean to First Nations Peoples?” and “If there were two things in the area of cancer survivorship that you would like to see done tomorrow—what would they be?” Yarning circle participants were given 20 min discussion time for each topic and at the end of each of these, all participants re-grouped for a larger group discussion on the topic.

The online Yarning circles were audio and video recorded through Zoom [28]. Recordings were transcribed verbatim andchecked for accuracy by the research team (DL, BB). Researcher field notes were used to supplement the recordings where it was difficult to transcribe the discussions. One researcher (AdG) undertook inductive thematic analysis according to Braun and Clarke’s process [29]. Transcripts were comprehensively read numerous times and recordings viewed and listened to until the researcher was immersed and familiar with the whole dataset [29]. The researcher recorded initial responses to the data in written memos [29]. Transcripts were colour-coded to denote expert types and ensure that First Nations’ voices were privileged throughout the analysis process [19]. Coding was undertaken using NVivo 12 software and, as no a priori codes were defined, codes were generated in response to the data [29,30]. Throughout the coding process, further written memos recorded researcher insights, responses to the data, and observations of discourse [29,31]. For some data extracts, video and audio recordings were reviewed to ensure coding appropriately represented meaning. In these cases, codes were confirmed in discussion with the research team (DL, BB, AH).

After initial coding, all codes were revised through a non-linear iterative process and collated into themes and sub-themes. Theme significance was determined by coding frequency and importance to the participants [29]. Themes were reviewed by returning to raw data to confirm internal and external heterogeneity (within and between themes, respectively) [29,32]. Prior to researcher consensus, participant checking was undertaken to confirm accuracy of results. Different theme trees iterated by AdG and discussed across the research team to explore theme constellations within the data and interconnections between concepts represented by themes [29]. In line with an Indigenist research approach, First Nations researchers (GG and NH) re-read the transcripts and provided a visual representation of the data. Final consensus of themes was undertaken by three Aboriginal researchers (AG, NH, GG).

Our research team recognizes the individual backgrounds, viewpoints, and values that each of us contribute to the project. The first author (AdG) is a second-generation immigrant woman from a rural area in Lutruwita (Tasmania) and was completing a Master of Public Health during this project. AG (Truwulway), NH (Palawa), and GG (Kamilaroi) are First Nations researchers with experience in conducting qualitative research with First Nations communities. BB, DL, and AH are non-First Nations researchers with experience in qualitative (BB; AH) and quantitative (DL) research. Throughout this study, AdG undertook reflexive practice to recognise how personal attributes might influence findings and to investigate relational accountability to the research participants, according to the Indigenist methodology [25,33]. An initial standpoint statement was developed to acknowledge biographical, social, and value-based assumptions before AdG reviewed the data [34,35,36,37]. A second standpoint, following theme description, aimed to reflect on assumptions and learnings that occurred or became apparent during the analysis process, and to understand how the research process impacted the researcher [35]. The results of this reflexive practice are included in the strengths and limitations of this study.

The study was approved by the Human Research Ethics Committee of Northern territory Department of Health and Menzies School of Health Research (2021-3995) and the University of Queensland Human Research Ethics Committee (2021/HE002245).

## 3. Results

Forty-four individuals agreed to participate in this study, with thirty-eight participants attending on the day. Participants self-identified as First Nations people living with and after cancer (*n* = 11), health professionals (*n* = 15), and researchers or policy makers (*n* = 12). Of the total sample, 37% identified as First Nations and 79% identified as female.

Our analysis revealed how themes derived from the Yarning circle data were interrelated, indicating the holistic nature of First Nations Peoples’ experience of living well with and after cancer. The three interconnecting themes (hope, culture, and family) were fundamental aspects of living with and after cancer and were more appropriately represented circularly rather than linearly. Therefore, in our representation of the Living Well With and After Cancer Wheel (Figure 1), the person living well with and after cancer is at the centre of the wheel and is surrounded by their family and culture, and hope. Each spoke of the wheel represents a priority area: strength-based understanding of cancer, cancer information, access to healthcare and support, and holistic cancer services. If one spoke of the wheel becomes weak then the entire experience of the person living with and after cancer may be affected. This connectivity occurs throughout the cancer experience.

Questions were raised around the appropriateness of the terms survivor and survivorship according to First Nations people, with one participant asking “*Am I a survivor? I am still living it*” (FS7). Although there was no consensus on preferred terminology, further investigation was suggested to “*ask a few people what they think of [the wording of it]*” (FS1). These results will be reported with preference for the term living well with and after cancer. Throughout this section, quotations in italics indicate voices of First Nations participants, while quotations from non-First Nations participants remain unitalicized.

### 3.1. The Living Well with and after Cancer Wheel

#### 3.1.1. Hope

The entire wheel is underpinned by the hope generated when the four spokes of the wheel are strong and the person and their family experience ongoing access to healthcare and support, strength-based understanding of cancer, cancer information, and holistic cancer services, which is evident throughout all themes below. Hope has been associated with improved cancer survival outcomes for many populations, and reduced distress and burden of symptoms when living with and after cancer [38]. Each theme forming a spoke of the wheel expresses elements of hope relating to the experience of living well with and after cancer. Participants expressed the need for hope to survive cancer, including communicating with communities about having “*a big chance to survive*” (MS1), as described further within Section 3.2.1 (Strength-Based Understanding of Cancer) below. Similarly, Section 3.2.2 (Cancer Information: What You Need to Know) describes the need for cancer information to be provided in a meaningful manner, alongside advocacy and healthcare navigation support, so that people do not lose hope and “… *fall through the gaps, really quickly and become quite disengaged …*” (FP3). In Section 3.2.3 (Access to Healthcare and Support), participants explain how “*people would just give up*” (FS7) due to overwhelming effort and financial strain while living with and after cancer unless they can access the support needed to maintain hope. Finally, Section 3.2.4 (Holistic Cancer Services) describes the need to “*actually start to put black faces into cancer services*” (FP3) to provide quality and culturally safe cancer care services that meet the holistic health and wellbeing needs of First Nations people living well with and after cancer.

#### 3.1.2. Respect for Culture

Respect for culture is vital to the integrity of the wheel and the person’s cancer journey. For non-First Nations healthcare professionals to build a meaningful connection with First Nations people living with cancer, they first need to understand the critical importance of culture. Having respect for and acknowledging culture as central to many First Nations people living with cancer is imperative to providing culturally safe services and support that will see First Nations people live well with and after cancer. It is important to note that culture exists on a continuum, so First Nations people with cancer will not necessarily have the same cultural values or experience culture in the same way [39]. Therefore, appreciating the diversity of cultural expressions and needs amongst First Nations Peoples is necessary for the delivery of culturally respectful cancer care.

An important part of First Nations Peoples’ culture is its collectivist underpinnings that emphasise the importance of the collective over the individual. This is particularly relevant when considering the impact that an individual’s family and community may have on their cancer journey. One First Nations healthcare professional explained that “…*[people living with cancer] are so tied up with worrying about family and community back home, half of their focus is on that and not even on their own survival*” (FH4). Another participant exemplified this with a story about his sister, whom he thought “*should have listened to me but she wouldn’t, she had to go to the bloody wedding instead of getting treatment when she was diagnosed*” (MS1). Non-First Nations healthcare professionals do not always understand the fundamental cultural values outlined above, as one participant explained: “… *we noticed here with our services—they don’t realise that our Mob don’t think about our own individual bodies*” (FH4). However, this understanding may be necessary to establish respectful therapeutic relationships through which honest cancer conversations can take place.

Diverse cultural ways of Knowing, Being, and Doing shape the experience of First Nations people living with cancer but may be difficult for non-First Nations healthcare providers to understand. First Nations Peoples comprise “… *one people but many different cultures and beliefs. So, when you talk about significant cultural awareness you’ve got to come up with some sort of solution or treatment or plan that can be adapted to suit current situations when you come upon them*” (MS3). One First Nations woman described cultural awareness required by healthcare professionals to facilitate health communication: “… *so that when [First Nations people go] even just into the hospital, the health staff there are knowledgeable in some of the traditions that are practiced. Men can’t look women in the eye, or vice versa, where women don’t look men in the eye*” (FS2). This participant continued, illustrating the nature of cultural barriers: “*So how do they make a consultation when their head is down and they think they are being ignored, when they are trying to pass that information on?*” (FS2). Participants described how cultural protocols may be required for some First Nations people but not others, indicating the need for healthcare professionals “… to know [patients] as an individual; who am I, what’s my situation, who’s my family, what are my needs… getting to know the person is the most important element of it” (FR3). These findings indicate the importance of cultivating cultural respect tailored to the values and beliefs of the First Nations person and their family to support meaningful therapeutic relationships in healthcare environments.

Alongside the traditions described above, diverse cultural needs often relate to Country, spirituality, food, and traditional medicine. One participant mused on the sense of identity they gain from living on Country: “*I’m up in [Place] Country now and as far as I’m concerned that’s a part of who I am*” (MS1). Another discussed the role of spirituality as “… *underpinning all of the health diagnoses that you might have; it’s how your spirit is travelling as to how you might fit along [a health and wellbeing] continuum*” (FS6). Several participants reflected on good food to promote wellbeing during cancer treatment. A First Nations man explained being in hospital without “… *the right sort of food and it was terrible. One of the things about healing—you get the right sorts of food*” (MS4). Similarly, a non-First Nations healthcare worker described being helpless to provide usual food for a patient who had travelled to the city for cancer treatment, asking “‘What would you eat if you were at home?’ and she said ‘Turtle’ and I went ‘mm … turtle’s not going to be so easy’ and it was the sign that we … were just not on the same page in ability to help her” (FH6). Another participant noted the role of “… those things from traditional medical practice … for things like spiritual wellbeing” (MR2). These discussions highlight the importance of acknowledging diversity through cancer care that is highly responsive and able to adapt to respect a range of cultural needs.

As well as the healthcare interactions described above, post-treatment support services must also be provided in a culturally safe manner. One First Nations woman noted that “… *there’s such a lack of culturally appropriate support services*” (FS6). Another confirmed how both support and cancer information “*needs to be culturally safe*” (FS1). A policy maker, detailing how to establish cultural safety, suggested “*embedding cultural understandings*” in post-treatment services for First Nations people living with and after cancer, “*include[ing] collective understandings and values as well as ensuring it is in the correct language and has effective communication*” (FP4). Culturally safe approaches to post-treatment cancer care may therefore embody values and understandings that diverge from those inherent in mainstream services based on Western ideas of quality of life and wellbeing.

#### 3.1.3. Importance of Family

Family for First Nations Peoples includes extended family as opposed to the nuclear or more immediate family structure more common in non-First Nations communities. Care addresses individuals’ needs throughout the cancer journey, yet First Nations participants spoke of living well with and after cancer in the context of family. Several participants described how cancer happens not only to the individual, but to “*the whole family. Your whole family is part of the healing process*” (MS4). In turn, participants explained how their cancer experience impacts their family. A First Nations woman described “*thinking how [having a double mastectomy] will impact on my family too. … they want me to do it, because they will feel more comfortable about my cancer risk in the future*” (FS4). Provision of cancer care and support should move beyond person-centred care to deliver services that cater for the interrelationships between individuals and their family members.

Family provides essential support, such as contributing to health literacy and helping individuals make informed choices around their cancer. One First Nations man described how family “… *are going to be your primary care when all this is done and you’re at home doing all the recovering and the recuperation*” (MS3). He continued to explain why family members need to understand information throughout the experience of living with and after cancer, because “… *if they’re not there with you as you’re going at each of those steps, they can’t help make informed decisions with you and support you*” (MS3). The medical terminology comprising cancer information is complex for individuals to relay if family are not present at consultations. As one participant explained, “… *even though I work for a medical service … and it’s easy to explain to us, but that information, it’s just hard to take home and explain to families and friends*” (FS2). Participants emphasise how accurate cancer information “… about the disease itself, the outcomes and so on…” (MR2) needs to be communicated not only to individuals but also their family members. These insights indicate the need to include family when sharing health information at cancer consultations.

Post-treatment support services may be required to address family and individual needs. Several participants described how individuals and their family members living with cancer require support to manage stress and fear associated with cancer. One participant considered their family’s experience of cancer, describing “*The stress when you go for your annual check-up—it’s the stress for me and for my family. They’re not really involved in that, but they should be … involved in that process … They don’t necessarily talk about [cancer]. But there need to be avenues for them to talk about it*” (FS4). In future, post-treatment support needs for the individual should also include the needs of family members.

### 3.2. The Spokes of the Wheel

#### 3.2.1. Strength-Based Understanding of Cancer

Participants described the importance of building a strength-based understanding of cancer as a chronic disease that can be managed and lived through, rather than a death sentence. Across Yarning circles, there was agreement on the need for increased understanding of hope to live well with and after cancer. First Nations participants suggested sharing positive stories of living with cancer, such as “… *promote[ing] the ones that have survived it for, say, 20 or 30 years … to tell [our own cultural Mob] there is a big chance to survive*” (MS1). One healthcare professional suggested that there are not “… very many good news stories, and so [with] the diagnosis of cancer automatically people think that they’re going to die” (FH7). Another healthcare professional discussed disseminating stories illustrating the experience of living with cancer “… so we will be able to say, ‘Well actually people do survive cancer but this is what it feels like’” (FH1). A First Nations woman further framed a strength-based approach as “… *celebrating cancer … we use the term ‘champions’ a bit at work* …” (FS6). These findings indicate that by promoting positive stories of cancer experiences, there is an opportunity to improve understanding of cancer information and generate hope of living well with and after cancer among First Nations people.

In addition to addressing normalised fatalistic beliefs regarding cancer, the sharing of stories from those living well with and after cancer were suggested to increase knowledge around this experience and provide peer support. First Nations participants conveyed the value of sharing cancer stories to both build connection and support between peers and to relay detailed information about living well with and after cancer. Several First Nations women requested to be linked with “… *a survivor buddy, someone who has been through the same experience and can come and talk with people, someone in the same kind of situation, who has survived … to say their experience and what you can do, and these are the resources available, and this is what I did*” (FS1). One stated the need for “… *sharing stories and I think creating support networks or support groups for Aboriginal people, personally I would have loved that when I was going through—even now that I’ve finished treatment, I’d love to sit and talk to other Aboriginal people that have gone through the same thing*” (FS6). Another explained how “… *the talking circles are just so important for community*” (FS7). These findings demonstrate the value of incorporating storytelling into cancer support services to enable sharing of cancer experiences as a means to build supportive environments and facilitate connections.

#### 3.2.2. Cancer Information: What You Need to Know

Gaps in cancer knowledge span the experience of living with and after cancer. Participants highlighted a lack of understanding around post-treatment needs, amongst both healthcare providers and First Nations people living with and after cancer. Meaningful information relating to treatment effects and outcomes can also be communicated more effectively to enable informed decisions around healthcare options. As abovementioned, cancer knowledge may be shared through storytelling as well as through positive interactions with healthcare providers and accessible cancer resources. In addition to accessing cancer information, First Nations people living with cancer require high levels of health literacy and self-advocacy to navigate the healthcare services usually required to live well with and after cancer.

Participants specifically articulated the need to increase awareness around living well with and after cancer during the post-treatment phase. One First Nations participant explained how “*… there is a lack of action after treatment has finished, and I think that that’s a worry in itself*” (FS6). A healthcare worker described how people “… go back to the community [after treatment], … they do not understand that there are important checks to be done or there are important healthy living messages or practices to be useful for them during survivorship” (MH2). Healthcare providers should meaningfully communicate vital post-treatment messages to individuals and their families and understand the access barriers some First Nations people experience to a range of services: “… instead of just finishing treatment and the cancer’s gone, but actually there’s been all of these things that we’ve done and it takes time to recover and that people need support with that.” (FR4). However, healthcare providers themselves may not assess and communicate post-treatment needs: “… there’s no proper assessment … where we actually inform the clients of client services available … the follow-up care that’s required, and also inform the community health centres … given the health practitioners do not understand the outcome of the treatment in their community” (MH2). An improved model of care is required, whereby tertiary and primary healthcare providers understand how to appropriately communicate post-treatment information and support for patients and their families.

Alongside understanding of post-treatment needs, increased information is also required around treatment access and outcomes, particularly to facilitate informed decisions around treatment options. One First Nations researcher specified the need for more information to enable access to cancer treatment, including “… *early detection and symptom awareness and where to go for help, where to go, who to see, who to talk to*” (FR6). A First Nations woman described how cultural barriers may impair healthcare communication sharing the knowledge necessary for informed decision making. This participant stated that often interactions are not: “*… culturally appropriate, and because of [health professionals’] lack of understanding, [First Nations patients] don’t understand what treatment is for, and whether you have that choice whether to have that treatment or not … Like, it could be harmful to you, you could end up with this, you could end up with that.*” (FS2). This speaks to the importance of not only ensuring that patients are provided with the information they need to make informed decisions across the cancer continuum, but rather the vital importance of the communicative environment in which this information is provided.

When cancer information is communicated in an effective way, alongside a respectful patient–clinician therapeutic relationship, this interaction is seen in a more positive light, as one First Nations woman describes: “… *a good relationship with my doctor, she was very supportive and I was savvy on things, but there are a lot of people who aren’t*” (FS1). In contrast, a First Nations healthcare worker suggested how prevalent use of Western biomedical language and medical jargon may reduce treatment awareness: “*Sometimes… they don’t understand the medications and the importance, because with reading the directions*” (FH4). Similarly, another healthcare worker discussed sharing post-treatment cancer information according to literacy level: “… the explanation of ‘there is no cancer in your body but there is a probability of it coming back’ … those types of concepts need to be better framed for Aboriginal people in terms of the different literacy level” (MH2). A First Nations man emphasised the need to make cancer resources accessible “… *in picture forms, because the ones that can’t read and write need to be able to see things too … having the big words attached to them, it’s just not there*” (MS1). Information delivered through healthcare interactions and cancer resources should be appropriately developed to be meaningfully communicated and respect First Nations Ways of Knowing, Being, and Doing.

Beyond understanding information about the cancer journey, health literacy and advocacy are also necessary to navigate the health system and access cancer treatment and services. A First Nations participant emphasised the difficulty of navigating cancer care for people without “… *a knowledge or awareness of the hospital or healthcare system because it’s really complicated. There’s really not anyone there to support you* … *it has to be about me, so person centred and then yeah, just someone else to help me navigate and coordinate*” (FS5). Another linked the complexity of the healthcare system with reduced health services engagement, describing how “… *people really fall through the gaps, really quickly and become quite disengaged … Those that have survived have really taken control of their survivorship really forced you know treatment … I think that’s been their key to their survival*” (FP3). A healthcare worker promoted advocacy and navigation support for patients requiring treatment, “… talking about advocacy in the community and then when the patient comes into the acute setting, that there’s someone here to help them navigate the health system … guide them through that acute part of the journey where they’re just so overwhelmed” (FH7). A First Nations woman specifically suggested the importance of advocacy for rural communities, stating “… [I] *believe that advocacy should be available to our people in our communities*…” (FS2). As well as building personal skills and knowledge amongst First Nations people living with and after cancer, the provision of quality services is critical to address health system navigation and advocacy needs.

#### 3.2.3. Access to Healthcare and Support

As described in previous sections, access to cancer services is impacted by understanding cancer information as well as the health literacy and advocacy required to navigate the healthcare system. Access is also affected by availability and affordability of services, particularly for First Nations people living in remote areas. Availability gaps exist broadly, with a particular need for post-treatment support services in remote areas. Affordability barriers of healthcare and travel costs continue to burden ongoing everyday finances for First Nations people living with and after cancer. As discussed in the following section, primary healthcare may play an important role in addressing this post-treatment care gap.

Participants discussed the need for more post-treatment services in general as well as specifying a need for locally-based services in remote areas. One First Nations woman discussed service gaps in remote areas, stating: “*Our people in [remote] communities do not have, are not fortunate enough to have some of the services we have here in [regional centre]. Means they have to travel*” (FS2). A First Nations man highlighted lack of post-treatment services, explaining how “*it’s from when they come home from the treatment that they need that backup support and when our Mob is not getting it*” (MS2). Another First Nations participant confirmed this, saying “*… there just wasn’t any [support] available. So just sent away to hospital, and then sent back home.*” (FS1). Primary healthcare services were discussed in provision of support services: “there has got to be pathways to have support if you’re in remote areas … what is the place [primary care] are going to take in a survivorship model of sustainable care that allows people to be treated where they are” (MR2). A possible role for primary healthcare in coordinating post-treatment support was suggested and is discussed in the following section (Section 3.2.4. Holistic Cancer Services).

Many participants discussed how cost and travel challenges may reduce access to cancer care services. One First Nations woman stated: “*It is really expensive. … I’ve been thinking a lot about how much time and effort and finances it has taken to get me through that first 10 years. And I can see people would just give up, because there was a lot of it*” (FS7). This perspective was detailed further by another First Nations participant explaining how “*[people] think ‘I know that I can’t afford to access a clinician because of costs and that I have to travel and those things’ and so they don’t even bother getting screening*” (FP4). The costs of travelling to access cancer care arise due to “*the distance and transport and accommodation, I think they’re things that we do worry about because of the finance side of things*” (MS1). Accessing cancer care away from home may interfere with employment, further exacerbating the financial burden of cancer over time. One participant explained how this impacted on her and her family where varying cancer treatments required her to travel to different locations: “*…for me to travel too, because I work as well, that put me out with my family as well as work*” (FS2). Improvements to geographic and financial support may be required to improve access to services required throughout the cancer journey.

Ongoing post-treatment costs of cancer care pose long-term financial challenges. As one healthcare professional explained, “[living with and after cancer] is about going back to the hospital again and again and again, and it’s about having more surgeries, and it’s about having more treatments, and more follow up. … to keep going for those visits … they’re expensive to attend” (FH10). A First Nations man detailed how these financial burdens are experienced, describing how “*[community members] were worried about the family, the food, the rent, the cost, who’s gonna pay the mobile phone. The ongoing costs.*” (MS4). Similarly, a healthcare professional shared the story of a “… gentleman who’s from a remote community who’s been prescribed [a supplement]. But he can’t afford that. And that’s like $30 and he can’t afford it. So, we’re chucking money out of our own pocket to support him, to buy his food” (FH4). These findings highlight how the cost of ongoing cancer care may inhibit treatment access and adherence, which could contribute to reduced use of healthcare services amongst First Nations people with cancer [40]. Therefore, there is a need to develop policy and advocacy measures to minimise ongoing post-treatment healthcare costs and ease financial pressure, particularly for people who need to travel to access cancer care.

#### 3.2.4. Holistic Cancer Services

Quality, culturally safe, and coordinated cancer services are required to live well with and after cancer. Care coordination is required after cancer treatment to ensure that ongoing health and support needs are met. The role of primary healthcare was discussed in coordinating holistic post-treatment cancer care, potentially through a cancer wellness plan funded by the Medicare Benefits Schedule (MBS). A First Nations cancer-specific workforce was described as necessary for high quality and culturally safe cancer care services throughout the cancer journey. This workforce should be established in ways that build the capability of First Nations health staff.

Participants confirmed a lack of care coordination and navigation support when living with and after cancer, particularly following treatment. As one First Nations woman explained, “*I very much have to self-support and self-direct myself in regard to the cancer journey so there’s not a coordinated approach to my care following treatment. I have to do it*” (FS5). To address this need for integrated post-treatment care, a healthcare provider suggested a novel “… integrated supportive framework that needs to be introduced to all patients that finish treatment. So that is obviously not in our health system at the moment” (MH2). Another described how the scope of care coordination may encompass comorbidities as well as cancer to provide holistic ongoing healthcare: “… in such a way that you kind of coordinate mental health support as well as cardio-metabolic … as opposed to saying, ‘It’s Tuesday today; it’s the cancer day, but come back tomorrow so we can look at your heart’” (FH1). Care coordination delivered through primary healthcare may be an effective way to manage multiple chronic diseases, including cancer, in a holistic manner.

Participants suggested integrating post-treatment cancer care by clarifying the role of primary healthcare. As a healthcare worker proposed, “[the] cancer profile in primary care for survivorship should be on top of the agenda side-by-side with the other chronic diseases and then we can incorporate all the required training and all the resources required to address the issue” (MH2). Primary healthcare providers may be well-positioned to facilitate quality post-treatment care by delivering a holistic cancer wellness management plan that “… captures all the important elements of survivorship care in general …. be it in primary care at the community control level, or also importantly if the person is receiving treatment just in the mainstream primary care” (FP1). This plan was suggested to incorporate allied health and support services, as well as “*… things like your exercise, nutrition, regular check-ups*” (FS1). It was suggested that, where available, these services could be subsidised through a designated MBS item number. A healthcare worker suggested “[an MBS] item number for the work the GP will do to set [the wellness plan] up. That’s easy to change relatively. I mean, it’s a federal issue” (FH6). There is scope for a holistic model of post-treatment cancer care to address wellbeing needs associated with cancer as a chronic disease.

In order to deliver quality and culturally safe cancer care, many participants discussed the need to expand the First Nations cancer workforce. A First Nations participant stated that “*… none of this [cancer planning] can actually happen until we actually start to put black faces into cancer services*” (FP3). A First Nations woman explained the importance of training specialised First Nations cancer workers and not placing extra load on already overburdened First Nations health workers, explaining, “*We do need to have separately employed cancer care workers. [First Nations Health Workers] … shouldn’t be the main worker that works with that particular client … no one’s expecting [a renal nurse] to work in the cancer unit … or go over there and give advice. So the same applies to our Aboriginal Health Workers*” (FS7). Participants suggested that specialised First Nations cancer workers should receive training and recognition beyond what current First Nations health workers receive. A researcher described how “the issue with Aboriginal Liaison Officers is often they are overworked and undertrained, so I think [the specialised cancer worker role] needs to be an ongoing professional fully trained role that is respected by the healthcare providers as an equal partner. I’m not convinced that that is always the case with Aboriginal Liaison Officers” (FR1). Other participants promoted cancer-specific nurses working alongside First Nations health workers: “*having the Aboriginal liaison officer/breast care nurse model would work really well*” (FS5). This priority identified in the data suggests more cancer-specific nurses may be required in addition to current First Nations health worker roles.

Building a sustainable and effective First Nations cancer workforce will require robust recruitment, training, and ongoing financial support. As a First Nations participant proposed, “*… the only way that we can get Aboriginal nurses specifically into cancer, is to provide some supported orientation and employment processes*” (FP3). A healthcare professional reflected on sustainability across the entire cancer workforce, suggesting “There needs to be some flexibility about how you find [cancer nurses], and that you’re not just sucking workforce out of other areas” (FH6). Another participant added perspective on cancer workforce retention, pointing out that “There’s no point in asking for the workforce to be paid without … education and support because they’re tough jobs, particularly the metastatic ones” (FR1). Innovative initiatives may be required to engage an effective, empowered and sustainable First Nations cancer workforce.

## 4. Discussion

The results from this study present insights into the experience and needs of First Nations people living well with and after cancer. Despite unacceptable inequity in cancer outcomes and quality of life experienced between First Nations and non-Indigenous Australians living with and after cancer [9,17], the results of this study provide a way forward to improve healthcare services. Numerous ways to address this inequity are highlighted here, including cultural considerations for delivering person- and family-centred cancer services, integrating care throughout the cancer journey, promoting cancer awareness and support, and building the capacity and capability of a First Nations cancer-specific workforce. These findings thereby hold practicable relevance for increasing capacity across a broad range of stakeholders in First Nations cancer care provision, including First Nations individuals, families, communities, cancer service providers, healthcare workers, health service managers, and policy makers.

The First Nations person as an individual is placed at the centre of the Living Well With and After Cancer Wheel (Figure 1). National cancer care guidelines specific to First Nations Peoples similarly model person-centred care [41,42]. Yet, the results here indicate that for many First Nations people, individual needs only make sense within the family context in which they are nested. Therefore, in Figure 1 the family immediately encompasses the individual. This reflects the modelling of First Nations concepts of health and wellbeing, which also locate individual health as interwoven within family, community, and cultural contexts [14,15]. In this way, the results presented here support the redesign of current models of cancer care to more accurately reflect the key role of family in First Nations individuals’ lives. New models might include identifying, with the person diagnosed with cancer, key support figures in the family and community and how they would like to be included in planning and delivering care, including attending appointments and receiving healthcare communication.

This study confirms previous findings that First Nations people may prioritise family, community and cultural demands over their individual health needs [16]. Yet, the results from the current study further indicate that pursuing cancer treatment may not be a priority for First Nations Peoples unless family connections and opportunities to continue contributing to family and community life remain intact. This may have critical implications for the success of a range of cancer services in meeting the needs of First Nations people living with and after cancer. Internationally, there is increasing acknowledgement that varying health and wellbeing priorities often contribute to reducing the relevance of cancer care initiatives designed for Indigenous Peoples [12]. Indigenous Peoples’ perspectives tend to uphold collectivist values, defining health and wellbeing with greater holistic emphasis than individualistic approaches influencing Western health systems and biomedical practice [12,15]. Cancer Australia acknowledges that First Nations people may have cultural obligations and family and community responsibilities that may interfere with pursuing cancer care [41]. Each family has unique needs, and within family-centred models of care, cancer care providers could offer opportunities to address family needs in order to appropriately support the First Nations individual. However, there is currently limited information to guide how cancer care providers can respectfully support First Nations people navigating conflicting priorities when making health choices.

Person- and family-centred cancer care must also be provided in ways that respectfully address diverse cultural needs. Cancer Australia highlights the importance of acknowledging the diversity of First Nations cultures as a key element of cultural competence [41,43]. The current study also suggests that cancer care providers in a range of localities could understand the nuanced ways of Knowing, Being, and Doing specific to the people of each place, to effectively and appropriately deliver care. Previously, non-First Nations health staff have reported increased confidence in addressing specific cultural needs when working in health services with high proportions of First Nations health staff [44]. Similarly, highly successful cancer care provision for First Nations Peoples has been described, utilising two-way learning principles and First Nations staff leadership to build the capabilities of both First Nations and non-First Nations staff [44,45,46]. Locally oriented collaborative healthcare partnerships between First Nations and Non-First Nations staff may be required to deliver high quality person-centred care [46,47]. If based on the above principles, a cross-cultural collaboration model for health services may guide adaption towards culturally responsive care across Australia.

Health literacy support should be integrated throughout the experience of living with and after cancer, especially after initial cancer treatment. These findings confirm the need for health literacy support early in the cancer journey, to ensure health information is communicated in ways that enable informed consent and decisions about treatment [18,44,48,49,50]. This study also confirmed that early health literacy support may include community advocacy [49,51], culturally relevant cancer resources accessible for people with varied literacy levels [52,53], and care provided by either cancer-specific First Nations health workers or culturally respectful non-First Nations health workers [44,45]. However, these results further highlight the need to address post-treatment health literacy amongst First Nations people living with and after cancer. Furthermore, increased awareness, assessment, and coordination of post-treatment needs is also required within primary healthcare services to enable integrated care and support that continues as required through the experience of living with and after cancer. A model of care is required to determine how comprehensive support, including health literacy [49], community advocacy [49,51], health system navigation [47,54], care coordination [49,51,54], and financial support [18,53] can be facilitated in an integrated manner, along with tertiary and primary healthcare.

There is a pressing need for greater First Nations representation within healthcare settings, particularly in terms of a cancer-specific workforce. Numerous factors contribute to the under-representation of First Nations healthcare professionals, including collective experiences of grief and loss, racism, heavy workloads, and lack of support and mentoring [10,55,56]. Despite these challenges, the current findings confirm the vital role of First Nations health staff in providing culturally safe and high-quality care for First Nations people living with and after cancer [44,49,51]. Previous research has found a critical need for supported cancer-related First Nations health workers to address health system navigation and care coordination needs across primary and tertiary healthcare [49]. However, these results also indicate a need to train First Nations nurses specific to cancer types and support them to work sustainably in challenging cancer roles. There are national strategies that promote supportive workplaces that attract First Nations health staff and enable ongoing career development within culturally respectful environments [57]. In practice, health services that have implemented cultures of respect and First Nations leadership structures have been highly successful in engaging and retaining First Nations health staff as well as improving patient outcomes [45]. Future health planning should explore ways to implement similar organisational structures across mainstream health services to promote respectful employment for First Nations health staff nation-wide.

These findings reflect ways to increase hope and awareness around First Nations people living well with and after cancer. Cancer has historically been viewed, amongst many First Nations people, as a diagnosis without hope—that is, an expectation of premature mortality [52,58,59]. This study confirms the previously described need for sharing positive stories within communities to increase awareness around the possibility of living well with and after cancer [52]. Similarly, storytelling within a community cancer support network has helped dispel negative cancer beliefs and encouraged First Nations people to undertake cancer screening and engage with healthcare [60]. The current study showed that peer support networks that facilitate storytelling within community are important for building connection through shared experience. In addition, promotion of positive cancer stories was discussed at the community level and more broadly to increase widespread awareness of living well with and after cancer.

The findings from this study are strengthened by using Yarning methodology to honour First Nations Peoples’ Ways of Knowing, Being, and Doing and establishing an equal power balance between participants [19,20]. This promoted culturally safe sharing of stories, yet also enabled participants to discuss solutions in a conversational manner. The first author (non-Indigenous) followed close guidance from First Nations team members to address cultural blind spots and ensure methodological integrity and beneficial research outputs [21,61]. Additionally, as men’s experiences were under-represented in this research, recommendations relating to the delivery of First Nations men’s cancer care cannot be extrapolated in detail from this study. As First Nations men experience additional barriers when accessing healthcare and receiving a diagnosis, future research should focus specifically on priorities for this group following a cancer diagnosis. Furthermore, while several First Nations participants shared stories from rural communities, additional research is required to more fully represent the lived experiences of First Nations Peoples living with and after cancer in rural and remote areas.

## 5. Conclusions

These findings provide initial evidence on the lived experience and priorities of First Nations people living well with and after cancer. The findings presented here indicate the need for a model of integrated, family-centred cancer care to holistically support First Nations people throughout and beyond their cancer experience. Finally, health promotion resources created and distributed according to First Nations Ways of Knowing, Being, and Doing may be important for sharing cancer stories and increasing hope for living well with and after cancer. While these findings are directly relevant for First Nations Peoples in Australia, they may support development of similar models with First Nations Peoples around the world.

## Figures and Tables

**Figure 1 ijerph-21-00798-f001:**
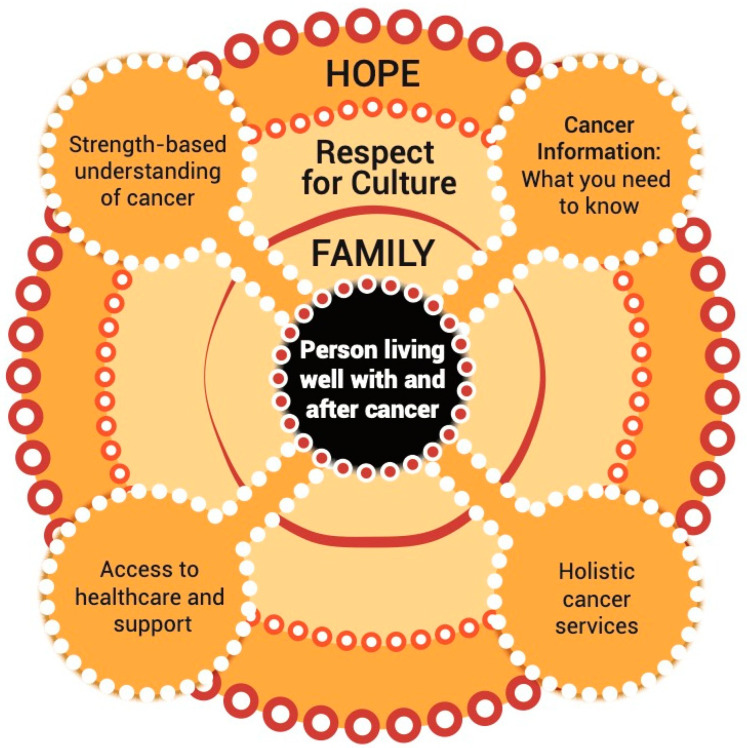
The Living Well With and After Cancer Wheel represents the interconnected and holistic nature of First Nations Peoples’ experience of living well with and after cancer.

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
