# Peer review of "‘How Your Spirit Is Travelling’—Understanding First Nations Peoples’ Experiences of Living Well with and after Cancer"

_ijerph, 2024, doi:10.3390/ijerph21060798_

Round 1

Reviewer 1 Report

Comments and Suggestions for Authors

The authors summarize their study of First Nations peoples' experiences of living well with and after cancer.  Data was collected through online Yarning circles and analyzed according to an inductive thematic approach.  The experiences of First Nations people living well with and after cancer diagnosis is inextricably connected with family.  Suggestions are made based on data collected in the research regarding integrating family-centred cancer care holistically to support First Nations people throughout and beyond their cancer journey.

The essay is well-written, well-organized, and well-researched.  

My only suggestion to the authors is this:  can you be even more specific about how programs and methods in cancer survivorship can change to meet the needs of First Nations people?  The more detailed the suggestions can be, the better that the programs and methods therein can change and respond to promote the health of First Nations people with cancer.  It is one thing to suggest the need to integrate holistic family-centred cancer care, and another to offer specific suggestions about how to do so. 

Author Response

Thank you so much for reviewing this article and providing valuable feedback requesting more specific detail regarding how programs and methods in cancer survivorship can change to meet the needs of First Nations people.

To address the point you raised, two sentences have been added to the discussion section, describing how family-centred models of care may be implemented. The first sentence is on lines 589-592 (previously 586), and the second is in the following paragraph, lines 606-608 (previously 600). Together, these two sections provide suggestions of how new models of care may key support figures in family and community to be included in care planning and delivery, and also how models might focus on meeting needs of the family to support the First Nations individual with cancer.

Please note, this research is the first stage in the development of a model of care, and subsequent publications will describe this family-centred care model in detail.

Reviewer 2 Report

Comments and Suggestions for Authors

I have very few comments to make on this manuscript - I consider that it makes a valuable contribution to our understandings of First Nations needs relating to living with and surviving cancer, - it is consistent with what I know of the experience here in Aotearoa, and adds some nice details.

The parts of the manuscript I've highlighted are where there are minor grammatical errors (apostrophes) or where quotes were not italicised (was this due to word length?). Because of the interrelatedness of the key aspects of the wheel, based on the themes, the results do feel slightly repetitive at times. In one instance I asked whether some content could be re-ordered. A careful checking over of the themes could ensure that repetition is minimised.

Author Response

Thank you very much for reviewing this article, and for providing such detailed and thoughtful feedback.

Please see the following points addressing the feedback you provided in the pdf file:

  • Line 58 - ‘received’ changed to ‘receive’
  • Lines 80-81 - removed ‘affected by cancer’ from the sentence ‘To date, there is limited understanding from the perspectives of First Nations peoples regarding principles, policies and service provision related to living well with and after cancer’ to reduce repetitive references to cancer.
  • Line 223 (previously 221) - ‘discusses need’ changed to ‘describes the need’
  • Line 302 - ‘individual’s needs’ changed to ‘individuals’ needs’
  • Lines previously 378, 383, 491, 510, 513, 519, 524, 545 and 560 (now 380, 385, 478, 513, 516, 522, 527, 548 and 563) - these quotes are not in italics as they are from non-First Nations participants, as per the explanation in line 203: ‘Throughout this section, quotes in italics indicate voices of First Nations people.’ For clarity, this sentence has been changed to ‘Throughout this section, quotes in italics indicate voices of First Nations participants, while quotes from non-First Nations participants remain unitalicized.’
  • Line previously 474 - You make a very good point, thank you. To clarify meaning, I have re-worded the sentence previously starting on this line (now line 493 - ‘Research has shown…’) to explain ‘These findings highlight how the cost of ongoing cancer care may inhibit treatment access and adherence, which could contribute to reduced use of healthcare services amongst First Nations people with cancer’
  • Line previously 482 - Thank you for this suggestion. The paragraph previously starting on line 482 has been moved so that it now starts on line 453. To accommodate for this change, wording has been changed on lines 446 to reflect the introduction of availability before affordability. Similarly, the order of the two sentences between lines 447 and 450 has been swapped.
  • Line previously 529 (now 531) - The acronym MBS is introduced in the first paragraph of this section - 3.2.4. Holistic Cancer Services

Thank you also for your suggestion that the results section may be slightly repetitive at times, and for acknowledging that this is due to the interrelatedness of the key aspects of the wheel. After reading through this section carefully several times, I have not been able to pinpoint sections that I would remove without losing some nuance of the experiences of participants. Therefore, without having any specific sections of the text to address, I have not made any changes based on this feedback, at this stage.

Reviewer 3 Report

Comments and Suggestions for Authors

Thank you for the opportunity to review your article, "'How your spirit is travelling' - Understanding First Nations Peoples’ experiences of living well with and after cancer." It was valuable to gain insights into the experiences and needs of First Nations peoples in cancer care.

The research provides a foundation for understanding the importance of culturally sensitive and responsive care. The proposed new model of care has the potential to significantly impact the well-being of First Nations peoples. Please consider a few suggestions below to improve the clarity to the larger audience.

 -suggest condensing a bit and avoiding repeated words like power imbalance in the following paragraph in the methods section: lines 92-105

“This study was guided by Indigenist methodology as a relational and transformative 91 approach to reflect First Nations peoples Ways of Knowing, Being and Doing [19, 20]. 92 Relational ontology locates the researchers and participants within a network of reciprocal 93 relationships, whereby each entity is attributed equal power and importance [19]. Trans-94 formative approaches examine underlying power imbalances affecting social, political 95 and historic contexts within which health is experienced by First Nations peoples living 96 well with and after cancer [19, 20]. In traditional research relationships, power imbalances 97 historically elevate researchers above participants [20]. With this methodology First Nations peoples' voices have been privileged [19, 21, 22]. 99

Yarning, an Indigenist research method, guided culturally safe, cooperative and con-100 versational data collection [23]. Yarning is an information-sharing method, whereby each 101 person is accountable for engaging and relating respectfully [19, 24, 25]. This method re-102 dresses power imbalance that can often occur between researcher and participant, ena-103 bling a safer environment for participants to share [24]. The current study included six 104 online Yarning circles [24, 26, 27]. Prompts were used to guide discussion according to the 105 research question [26].”

-the following might be included in the declaration section or end of the methods section

“Our research team recognizes the individual backgrounds, viewpoints, and values 107 that each of us contribute to the project. The first author (AdG) is a second-generation 108 immigrant woman from a rural area in Lutruwita (Tasmania) and was completing a Mas-109 ter of Public Health during this project. AG (Truwulway), NH (Palawa) and GG (Kamila-110 roi) are First Nations researchers with experience in conducting qualitative research with 111 First Nations communities. BB, DL and AH are non-First Nations researchers with experience in qualitative (BB; AH) and quantitative (DL) research.“

-please provide more detail about the participants. For example, in the paragraph below, it might be good to give more details about the type of healthcare professionals, researchers and policymakers
“First Nations peoples living with cancer and/or their family members (18 years of age 114 or older at the time of recruitment), healthcare professionals, researchers and policymakers were purposively recruited through investigator networks and the Clinical Oncological Society of Australia (COSA) Cancer Survivorship Special Interest Group to participate 117 in the virtual roundtable conducted in March 2021. ”

-Acknowledge the implications of less representation of males (21%) in research

-37% identified as First Nations, did the research aimed to present the views/results by First Nations and other Australians i.e. comparative analysis. If yes, it should be noted in the introduction and methods section. The results section should also adequately represent the quotes/voices of other Australians, given they comprise the majority of the sample size. The other option could be the study's focus on First Nations participants.

-please consider the more appropriate wording in some sections. For example, “must” in the following sentence could be replaced by other less strong words. The model is initial and needs more understanding to apply it “Health literacy support must be integrated throughout the experience of living with and after cancer, especially after initial cancer treatment.”

Author Response

Thank you very much for reviewing this article, and for providing such detailed and thoughtful feedback.

  • Your first comment refers to overuse of the term ‘power imbalance’ through 2 paragraphs in the methods section. To address this, the words ‘power and’ have been deleted from the sentence now reading ‘Relational ontology locates the researchers and participants within a network of reciprocal relationships, whereby each entity is attributed equal importance’ (lines 93-94). Additionally, the sentence starting on line 97 (‘In traditional research relationships, power imbalances...’) has been deleted, as this idea is repeated in the following paragraph describing Yarning research methods.
  • In response to your second comment, details about the research team now appear towards the end of the methods section, starting on line 163 rather than on line 106.
  • In your third comment you have requested additional details about participants. Detailed information about health professional, research or policymaker types was not collected as part of the yarning method. However, a new sentence has been added from line 110 - 112, describing how ‘The healthcare professionals, researchers and policy makers represented medical, nursing, and allied health professions, and all were involved in cancer care across state and national organisations.’
  • There was a brief acknowledgement of the under-representation of men in this research in the strengths and limitations paragraph. In response to your feedback, this has been expanded to include implications of this under-representation and future research recommendation, running from line 679 to 683
  • Thank you for asking whether this research aimed to undertake comparative analysis. This was not our aim, but rather we aimed to privilege, prioritise and amplify First Nations voices, contextualising the themes and narratives within the stories and experiences the First Nations participants shared. Additionally, the non-First Nations participants were part of the cancer workforce and didn’t necessarily have lived experience of cancer diagnosis but had experiences and understandings of supporting and making policies around First Nations people in cancer care. And so, comparison would not be appropriate for this research.
  • In response to your final comment, regarding the strength of wording (overuse of the word ‘must’). This wording has been amended in lines 70, 302, 309, 370, 383, 423, 615, 626 and 658 (e.g. line 309 - ‘must’ replaced with ‘should’, and line 370 ‘outcomes must also be communicated effectively’ changed to ‘outcomes can also be communicated more effectively’). In some instances, I have left stronger language to reflect the non-negotiable importance of culturally safe care provision (line 611)